# Comparative Performance of Serological (IgM/IgG) and Molecular Testing (RT-PCR) of COVID-19 in Three Private Universities in Cameroon during the Pandemic

**DOI:** 10.3390/v15020407

**Published:** 2023-01-31

**Authors:** Rodrigue Kamga Wouambo, Cecile Ingrid Djuikoué, Livo Forgu Esemu, Luc Aime Kagoue Simeni, Murielle Chantale Tchitchoua, Paule Dana Djouela Djoulako, Joseph Fokam, Madeleine Singwe-Ngandeu, Eitel Mpoudi Ngolé, Teke Apalata

**Affiliations:** 1Section of Hepatology, Department of Medicine II, University of Leipzig Medical Centre, 04103 Leipzig, Germany; 2American Society for Microbiology (ASM), ASM Cameroon, Bangangte, Cameroon; 3Faculty of Health Science, Université des Montagnes, Bangangte, Cameroon; 4Laboratory of Fundamental Virology, Centre for Research on Emerging and Reemerging Diseases (CREMER), Yaounde, Cameroon; 5Department of Biomedical Sciences, Faculty of Health Science, University of Buea, Buea, Cameroon; 6Department of Microbiology, Faculty of Health Science, University of Buea, Buea, Cameroon; 7Faculty of Medicine, Sorbonne University, 75013 Paris, France; 8Department of Medical Laboratory Sciences, Faculty of Health Science, University of Buea, Buea, Cameroon; 9Chantal BIYA International Reference Centre for Research on HIV/AIDS Prevention and Management (CIRCB), Yaoundé, Cameroon; 10Faculty of Medicine and Biomedical science, University of Yaounde 1, Yaounde, Cameroon; 11Faculty of Health Sciences & National Health Laboratory Services, Walter Sisulu University, Mthatha 5099, South Africa

**Keywords:** COVID-19, serological markers (IgM/IgG), prevalence, private universities, Cameroon

## Abstract

Background: COVID-19 remains a rapidly evolving and deadly pandemic worldwide. This necessitates the continuous assessment of existing diagnostic tools for a robust, up-to-date, and cost-effective pandemic response strategy. We sought to determine the infection rate (PCR-positivity) and degree of spread (IgM/IgG) of SARS-CoV-2 in three university settings in Cameroon Method: Study volunteers were recruited from November 2020 to July 2021 among COVID-19 non-vaccinated students in three Universities from two regions of Cameroon (West and Centre). Molecular testing was performed by RT-qPCR on nasopharyngeal swabs, and IgM/IgG antibodies in plasma were detected using the Abbott Panbio IgM/IgG rapid diagnostic test (RDT) at the Virology Laboratory of CREMER/IMPM/MINRESI. The molecular and serological profiles were compared, and *p* < 0.05 was considered statistically significant. Results: Amongst the 291 participants enrolled (mean age 22.59 ± 10.43 years), 19.59% (57/291) were symptomatic and 80.41% (234/291) were asymptomatic. The overall COVID-19 PCR-positivity rate was 21.31% (62/291), distributed as follows: 25.25% from UdM-Bangangte, 27.27% from ISSBA-Yaounde, and 5% from IUEs/INSAM-Yaounde. Women were more affected than men (28.76% [44/153] vs. 13.04% [18/138], *p* < 0.0007), and had higher seropositivity rates to IgM+/IgG+ (15.69% [24/153] vs. 7.25% [10/138], *p* < 0.01). Participants from Bangangté, the nomadic, and the “non-contact cases” primarily presented an active infection compared to those from Yaoundé (*p*= 0.05, *p* = 0.05, and *p* = 0.01, respectively). Overall IgG seropositivity (IgM−/IgG+ and IgM+/IgG+) was 24.4% (71/291). A proportion of 26.92% (7/26) presenting COVID-19 IgM+/IgG− had negative PCR vs. 73.08% (19/26) with positive PCR, *p* < 0.0001. Furthermore, 17.65% (6/34) with COVID-19 IgM+/IgG+ had a negative PCR as compared to 82.35% with a positive PCR (28/34), *p* < 0.0001. Lastly, 7.22% (14/194) with IgM−/IgG− had a positive PCR. Conclusion: This study calls for a rapid preparedness and response strategy in higher institutes in the case of any future pathogen with pandemic or epidemic potential. The observed disparity between IgG/IgM and the viral profile supports prioritizing assays targeting the virus (nucleic acid or antigen) for diagnosis and antibody screening for sero-surveys.

## 1. Introduction

Since March 2020, the world is facing a biological threat caused by the emergence of a new virus: Severe Acute Respiratory Syndrome-Coronavirus 2 (SARS-CoV-2) [1]. Named by the International Committee on Taxonomy of Viruses (CITV), SARS-CoV-2 is a 30 kb enveloped virus with a helical capsid whose genome consists of single-stranded, non-segmented, positive-polarity ribonucleic acid (RNA) [2]. It has four essential structural proteins: A spike surface protein (S), an envelope protein (E), a membrane protein (M), and a nucleocapsid protein (N) [3]. To usurp the human organism, its spike protein (S) binds via affinity and avidity forces to cellular angiotensin-converting enzyme 2 (ACE 2) receptors primarily expressed by respiratory epithelial cells from the nasal mucosa and secondarily by type 2 pneumocytes, hence its tropism for the respiratory tract and thus the preferential pulmonary involvement where SARS-CoV-2 causes emerging and potentially lethal atypical pneumonitis [4].

Since its appearance, four major waves of SARS-CoV-2 have been experienced [5]. As of 27 November 2022, the world has recorded 637 million confirmed cases and 6.6 million deaths globally [5]. By large, the United States of America (USA) is the most affected country in the world with over 98,972,375 cases [6]. In Africa, COVID-19 has affected all 47 African Region countries with 8,887,814 cumulative cases, which represented approximately two percent of the infections around the world [7]. South Africa is the most drastically affected country, with more than 3.6 million infections, followed by Cameroon with 123,993 cases of COVID-19, of which 1965 died and over 121,873 were recovered [7].

Many aspects of the COVID-19 pandemic remain unknown as it is asymptomatic in approximately 50% of cases where the subject recovers spontaneously (in acute or moderate forms) [8]. Nevertheless, in these acute forms, symptoms such as cough, moderate fever, asthenia, headache, and loss of taste and/or smell may be noted [9]. In addition, in the absence of a cure, the contamination of the subject evolves into an infection characterized by the appearance of symptoms. These symptoms appear progressively and correlate with the severity of the SARS-CoV-2 infection [9]. This severe form would be the result of a particular exaggerated inflammatory reaction characterized by a cytokine storm [3]. Subjects of a younger age are described as less likely to develop severe COVID-19 forms than adults.

The transmission of SARS-CoV-2 in the young depends on the local transmission rates, the circulating variants, the epidemiology of COVID-19 among children, adolescents, and adults, vaccine coverage for those eligible, and mitigation measures in place to prevent transmission [10]. At the heart of the COVID-19 pandemic, population restriction measures, including school closures and the introduction of barrier measures, were practiced worldwide to curb the spread of the pandemic [11]. Some evidence indicates that SARS-CoV-2 might spread more easily within high school settings than in elementary school settings [12,13,14,15], suggesting that SARS-CoV-2 transmission among children and adolescents is relatively rare, particularly when prevention strategies were in place [16]. However, close contact with persons with COVID-19, attending gatherings, and having visitors at home can increase its transmission rate [17]. In Cameroon, epidemiological data on COVID-19 infection in elementary and high schools are rare. To limit the spread of the pandemic, easy access to testing is important. However, the available diagnostic tools have not been used in many settings. This study sought to determine the infection rate (PCR positivity) and spread (IgM/IgG) of SARS-CoV-2 in settlements around three Cameroonian universities and suggest a suitable diagnostic algorithm for SARS-CoV-2 therein.

## 2. Materials and Methods

A.Study Design and Population

A multicenter cross-sectional study was conducted from 28 September 2020 to 6 September 2021 among 291 students in three private universities from two regions of Cameroon: Université des Montagnes (UdM-Bangangte) located in Bangangté, in the west region; the Higher Institute of Biological and Applied Sciences (ISSBA-Yaounde) and the Estuary Academy and Strategic Institute (IUEs/INSAM-Yaounde), both located in Yaounde, in the central region. Several reasons guided the choice of these two towns (regions): (i) Both are the most populated and highly heterogeneous cities in their respective regions; (ii) Yaounde, the capital city of Cameroon, is bigger than Bangangte and is equipped with an international airport of entry of foreigners; (iii) Yaounde was one of the starting points of COVID-19 and thus was among the first sites endorsed by the Ministry of Public Health for molecular diagnostic and management of COVID–19 cases; (iv) unlike Bangangte, Yaounde was essentially among the two first sites of multifunctional reference laboratories for SARS-CoV-2 molecular detection in Cameroon before the extension to other cities across the country thereafter.

After obtaining the required administrative authorizations, we included all students aged >18 years old from those three private universities (symptomatic and/or not) who signed the informed consent. Conversely, patients who declined the invitation and refused to sign the informed consent form were excluded. Additionally, those admitted to intensive-care units and individuals who declined either a blood stick or a nasopharyngeal swab were also excluded. From each participant, blood was collected by a prick on the middle finger and nasopharyngeal swab for SARS-CoV-2 IgM/IgG antibodies and RNA detection, respectively. The protocol of sample collection was performed according to WHO guidelines for COVID-19 sample collection. Samples were transferred using a cooler at +4 °C to the Center for Research on emerging and reemerging Diseases (CREMER).

An individual was considered symptomatic when he/she had at least three COVID-19-related symptoms such as taste disorders, loss of smell, fever, dry cough, breathing/shortness of breath, chest pain/pressure, aches and pains, sore throat, diarrhea, conjunctivitis, or headache (https://www.who.int/health-topics/coronavirus#tab=tab_3). The study aim was aptly explained to all the participants, and those who agreed to participate were recruited consecutively and completed the structured questionnaire with a member of the research team. Information on participants’ sex, age, university, history of COVID-19 symptoms, treatment, and comorbidity was collected. 

B.Ethics Approval and Consent

The research proposal was evaluated, and ethical clearance was obtained from the Institutional Review Board (IRB) of the National Ethics Committee for Human Health Research (N°: 2020/05/1218/CE/CRERSHC/SP from 6 May 2020). Additionally, we obtained authorization from the directors of the three Universities selected. An information note was given to all the eligible participants, who then provided their written informed consent before enrollment into the study. The confidentiality of study participants was secured via the use of identification codes.

C.Determination of Minimum Sample Size

The minimum sample size was obtained using the standard formula:
n = z^2^ × p × (1 − p)/m^2^
where “z” = the standard deviation of 1.96 (95% confidence interval); “p” = seroprevalence of SARS-CoV-2 antibodies found in Cameroon (IgM: 20% and IgG: 24%) [9], “m” = the degree of precision (0.05), and “n” = the minimum sample size.

D.Sample Collection and Conservation

Eligible participants who gave their approval were subjected to the collection and blood and nasopharyngeal fluid.

Blood Sampling

Blood was collected according to the aseptic and barrier measures of COVID-19. One drop of whole blood was required for the “on-site” testing according to the SOP presented to us by the “Abbott PanbioTMCOVID-19 “rapid diagnostic test kit (Panbio COVID-19 IgG/IgM Rapid Test|Point-of-Care—Abbott (globalpointofcare.abbott)).

2.Nasopharyngeal sampling

Safety procedures were performed according to procedural references for safe nasopharyngeal swab collection previously described [18,19]. Samples collected on nasopharyngeal swabs were moved to collection tubes containing a viral transport medium by breaking the swab at the groove. The sealed state and labeling were checked, and surface disinfection was performed [20].

3.Samples Transportation

Antibodies testing was performed on-site, whereas nasopharyngeal swabs were transported to CREMER, Yaounde, for diagnosis. All viral transport mediums containing nasopharyngeal swabs were temporarily stored between 2 and 8 °C immediately after collection. Nasopharyngeal swabs collected at ISSBA-Yaounde and IUEs/INSAM in Yaounde were directly transferred to CREMER and analyzed within 48 h. At the other sites, samples collected in Banganté were firstly frozen (−20 °C) in the Laboratory of UdM-Bangangte, then transferred once per week between 2 and 8 °C to the virology laboratory CREMER located in Yaounde (~400 km away) where they were analyzed upon arrival.

4.COVID-19 Testing

SARS-CoV-2 IgG/IgM antibodies were detected in whole blood using the Panbio™ COVID-19 IgG/IgM Rapid Test Device (REF: ICO-T40203, LOT: COV0052132, Expiration: 30 April 2021), according to the manufacturer’s instructions. The Panbio™ COVID-19 IgG/IgM Rapid Test is an immunochromatographic lateral-flow test kit used for the qualitative detection of IgG and IgM antibodies to SARSCoV-2 in human serum, plasma, venous, and capillary whole blood. In this study, all Panbio™ COVID-19 IgG/IgM Rapid Tests were performed on-site using capillary whole blood. That technique is reported to have a specificity of 99.4% and a sensitivity of 98.2% [21].

Molecular diagnosis of SARS-CoV-2 was performed using real-time RT-qPCR [22]. Briefly, after heat inactivation (65 °C, 10 min) of nasopharyngeal samples as previously described [23], RNA was manually isolated by a column-based RNA purification Kit of the DaAn gene (DaAn Gene Co., Ltd., Guangzhou, China). Then, the viral genome was detected via retro transcriptase quantitative polymerase chain reaction (RT–qPCR) analysis of the RdRp, E, and N genes using a DaAn Gene^®^ kit (DaAn Gene Co., Ltd., Guangzhou, China). Amplification of the SARS-CoV-2 genes was performed on a Quant Studio™ 7 real time thermocycler (Applied Biosystems, Waltham, MA, USA). Cycling conditions were as follows: Reverse transcription (45 °C/15 s) followed by initial denaturation (95 °C/2 min) and 45 cycles of [denaturation (95 °C/15 s), annealing (60 °C/30 s), and extension (72 °C/60 s)]. The internal control was included in each amplification. The cycle threshold (Ct) value of RT–qPCR was used to determine viremia and classify patients (negative or positive) as per the manufacturer’s instructions [22]. The prevalence of SARS-CoV-2 infection was defined as the proportion of individuals with positive RT–qPCR.

E.Data Analysis

Analyses were performed using the software package Stat view 5.1 for Windows (SAS Institute Inc, Cary, NC, USA). The continuous variables are presented in terms of mean ± Standard deviation (Std) and categorical variables as the absolute number (proportion in %). The associations between SARS-CoV-2 positivity or the serological profile and demographic and clinical characteristics were investigated by Chi-Square (Pearson or for trend), Mann–Whitney, or Kruskal-Wallis tests as appropriate. Univariate and multivariate regression analyses were conducted to identify factors independently associated with the risk of SARS-CoV-2 infection and serological profile. Only participants with all information were included in the analysis. For all the analyses, the significance level was set at *p* < 0.05.

## 3. Results

### 3.1. Determination of Baseline Characteristics of the Study Population

A total of 291 participants in three universities were surveyed between 28 September 2020 and 6 September 2021 in Yaoundé and Bangangté. Males and females had similar representations in the study with a sex ratio of 1:1. The mean age of study participants was 22.59 years old [min 18, max 27] and the age range most represented was 21–24 years (41.24%). The majority of these students were from the city of Bangangté 68.04% while 31.96% were from Yaoundé. Based on clinical presentation, most of the participants were asymptomatic (80.41% vs. 19.59% symptomatic) with less comorbidity (4.13%). It appears that the majority of the population had not yet been in close contact with an infected person (80.76%). In addition, most were sedentary ^2^ (93.13% vs. 6.87% nomadic ^1^) and had not yet taken any drugs against COVID-19 (98.63%) (see Table 1).

### 3.2. Prevalence of SARS-CoV-2 in Students and Associated Risk Factors

The overall prevalence of SARS-CoV-2 (positive PCR) among students was 21.31%. However, the PCR positivity rates differed across universities as follows: UdM-Bangante: 25.25% (50/198), IUEs/INSAM-Yaounde: 5% (3/60), and ISSBA-Yaounde: 27.27% (9/33). The identification of risk factors (sociodemographic and clinical characteristics) by binary logistic regression showed that females were two-fold more affected by SARS-CoV-2 infection than males (28.76% vs. 13.04%; OR = 2.21, 95% CI = 1.5–4.5; *p* = 0.0007). Similarly, students in the city of Banganté were almost two times more affected by COVID-19 than students from Yaounde (25.25% vs. 12.9%; OR = 1.95, 95% CI = 1.2–3.8; *p* = 0.01) (See Table 2).

### 3.3. IgM/IgG Serological Profile and Their Associated Factors

From the rapid serological testing (IgM/IgG), we noticed an overall IgM (IgM+/IgG− and IgM+/IgG+) and IgG (IgM−/IgG+ and IgM+/IgG+) seropositivity of 20.62% (60/291) and 24.4% (71/291), respectively. Furthermore, 11.68% (34/291) and 8.93% (26/291) of our participants had typical progressive (IgM+/IgG+) and acute (IgM+/IgG−) infection profiles. In addition, more than half of our participants did not have previous exposure to SARS-CoV-2 (IgM− & IgG−) (66.67%, 194/291) (Figure 1). The identification of factors (socio-demographic and clinical predictors) associated with the serological profile showed that two times more females had the progressive (IgM+/IgG+) profile compared to males (15.69% vs. 7.25%; OR = 2, 95% CI = [1.09; 5.18]; *p* = 0.02).

Students from the city of Bangangté had the majority of active infections (IgM+ & IgG+) compared to those from Yaoundé (14.14% vs. 4.45%; OR = 3.5, 95% CI = [0.95; 5.98]; *p* = 0.05). Students reported as “case-contact” were the least affected by active infection (IgM+&IgG+) (7.14%, *p* = 0.05); on the other hand, nomads were more affected by active infection (20%, *p* = 0.01) than sedentary participants (Table 3).

### 3.4. IgM/IgG Serological Profile and SARS-CoV-2 RNA Detection

Comparing serological and molecular profiles, we found that a proportion of 26.92% (7/26) presenting COVID-19 IgM+/IgG− had a negative PCR compared to 73.08 % (19/26) for those with a positive PCR (*p* < 0.0001). Furthermore, 17.65% (6/34) of negative PCRs vs. 82.35% (28/34) of positive PCRs were found in students with COVID-19 IgM+/IgG+, (*p* < 0.0001). Lastly, 7.22% (14/194) with IgM−/IgG− had a positive PCR (see Table 4).

## 4. Discussion

During this study conducted in three private universities in Cameroon (UdM-Bangangte, IUEs/INSAM-Yaounde, and ISSBA-yaounde), 291 students were enrolled. The mean age was 22.59 ± 10.43 years old [min 18, max 27], females were predominant compared to males (52.58% vs. 47.42%), and the majority of the population resided in the city of Bangangté (68.04% vs. 31.96% in Yaounde). Approximately three-quarters of the overall study population was asymptomatic (80.41%). This high number of asymptomatic participants may be related to the young age of the study population (mean = 22 years). In fact, studies conducted worldwide have presented COVID-19 as an asymptomatic disease in more than 85% of young subjects [24].

The overall prevalence of COVID-19 via RT-qPCR was 21.31% (62/291), with 25.25% at UdM-Bangangte, 27.27% at ISSBA-Yaounde, and 5% at IUEs/INSAM-Yaounde private Universities. Similarly, students of Banganté had twice as many cases of COVID-19 as those in Yaoundé (25.25% vs. 12.90%, *p* =0.01). This disproportionate prevalence between private high schools in Yaounde and Bangante could be explained by the fact that Yaoundé was among the cities considered the starting point of the COVID-19 epidemic in Cameroon in early 2020 and, thus, the initial point of the anti-COVID response [25]. The spread of the pandemic through smaller cities such as Bangangte was amplified by their lower response capacity. The detection of SARS-CoV-2 in these two cities of Cameroon informs us of the importance of the continuous application of barrier measures to prevent the spread of SARS-CoV-2. Our findings portrayed women to be twice as affected as men (28.76% vs. 13.04%, *p* < 0.0007), which differed from several studies [26,27,28,29]. Studies reported that even when men and women have the same prevalence, men with COVID-19 are more at risk for worse outcomes and death, independent of age [30]. Furthermore, most of the COVID-19 cases confirmed by RT-qPCR were asymptomatic (22.65% vs. 15.79% symptomatic). No severe cases were reported in this study. This could be due to the relatively young age of the study population. However, there is evidence that asymptomatic SARS-CoV-2 carriers can transmit the virus [31].

We noted a predominance of IgG (24.40%) compared to IgM (20.62%). Similar trends have already been reported with higher IgG levels in the general population in the USA [32], Cameroon [7,33,34,35], Congo [36], India [37], and Brazil [38], as well as in students in Spain [39]. In fact, experience earned from the kinetics of the antibody response from other viral infections taught us that, unlike IgM, which appears in the acute phase of infection, IgG is the marker of chronic infection and should appear later on in greater amounts [40]. Only 12.71% of our participants presented an IgM−/IgG+ profile. The serological profile in this study should reflect recovering or post-infection immunity as a result of infection with SARS-CoV-2 since none of our participants were vaccinated.

Concerning symptomatic subjects, only 9 out of 57 (15.78%) were PCR positive whereas 31.57% (18/57) presented at least one COVID-19 IgM and or IgG antibody [31.57% (18/57)]. SARS-CoV-2 prevalence [41,42,43,44] and IgM/IgG antibodies detection [3,45,46] vary according to identified symptoms. COVID-19 has similar symptoms to flu, which makes it difficult to differentiate from other respiratory diseases [47,48,49]. Thus, there is a need for a standardized method to detect SARS-CoV-2 RNA such as RT-PCR for its diagnosis in clinical practice [50]. Furthermore, as compared to PCR, antibodies tests look for antibodies in the blood that fight the virus that causes COVID-19 [51]. The presence of antibodies such as IgM/IgG does not always signal infection as they can also be detected in the blood of people who have recovered from COVID-19 or people who have been vaccinated against COVID-19 [51]. That is why anti-SARS-CoV-2 IgM and IgG antibodies have been expected to be useful as complementary tests, in addition to RT-PCR, for the diagnosis of COVID-19 [45].

Four serological profiles were identified: 12.71% IgM−/IgG+, 11.68% IgM+/IgG+, 8.93% of IgM+/IgG−, and 66.67 % of IgM−/IgG−. Similar results have been reported in the Congo [35], the USA [31], and Cameroon [32,34] with sample sizes of 684, 368, 291, and 971, respectively. No association has been found between serological profiles and socio-demographic characteristics. Our results diverge from those from the Congo where hospitalizations due to COVID-19 correlated with immune response [36]. However, our findings are similar to other studies in Cameroon reporting this association [32,34]. The differences seen between Cameroon and the Congo may be accounted for by the difference in sample size and study context. In fact, the study in the Congo was performed on 684 travelers [52], and those in Cameroon by Nguwoh [32] and Nwosu [34] were conducted on 368 and 971 healthcare workers. Moreover, more than half of our study participants did not have previous exposure to COVID (IgM− & IgG−) 66.67% (194/291). This large proportion of non-exposure could reflect the success of the nationwide measures implemented by the Cameroonian government to stall COVID-19 transmission [24].

In this study, approximately one-quarter (26.92%) of the patients with serological markers of acute COVID-19 infection (IgM+/IgG−) and one-fifth (17.65%) with serological markers of ongoing COVID-19 infection (IgM+/IgG+) had a negative PCR result (*p* < 0.0001). The difference observed between antibodies and PCR is logical and supported by previous findings [4,16,34,53,54]. The observed disparity between serologic (IgG/IgM) and molecular (PCR) profiles could be accounted for by the acute nature of such an infection with an incubation time of 2–14 days, which differs from the peak production of IgM and IgG (~10–14 days) [4,53]. This further supports the use of PCR for diagnosis and antibodies for serosurveys.

Only 2.7% of the participants with serological markers of postinfection immunity (IgM−/IgG+) still had positive PCR results. In fact, IgM−IgG+ is the conventional serological profile of past exposure to infections or chronic infection, a potential indicator of a cure, or natural immunization [55]. Finally, 7.22% (14/194) of the participants negative for IgM− and IgG- had a positive PCR result. These may be false negatives or evidence that these are early infections with levels of antibodies that cannot be detected by the serological tests we used [36]. Moreover, it could be related to seroconversion, a period before the onset of antibody production, which will increase detectability [56,57,58]. In addition, numerous studies have already reported low sensitivity of COVID IgM/IgG serological tests compared to techniques such as ELISA or LUMINEX [59,60,61,62,63], suggesting a continuous readaptation of prefabricated RDTs in the emergency anti-COVID response is needed, especially as circulating strains arise and cause new waves.

## 5. Conclusions

During the COVID-19 pandemic, there was an upsurge (above 20%) of COVID-19 confirmed cases within the University settings of Cameroon, calling for increased preparedness and rapid response strategies in higher institutes in case of any new pathogen with pandemic or epidemic potential arises. The observed disparity between IgG/IgM and the viral profile supports prioritizing assays targeting the virus (nucleic acid or antigen) for diagnosis and antibody screening for sero-surveys.

## Figures and Tables

**Figure 1 viruses-15-00407-f001:**
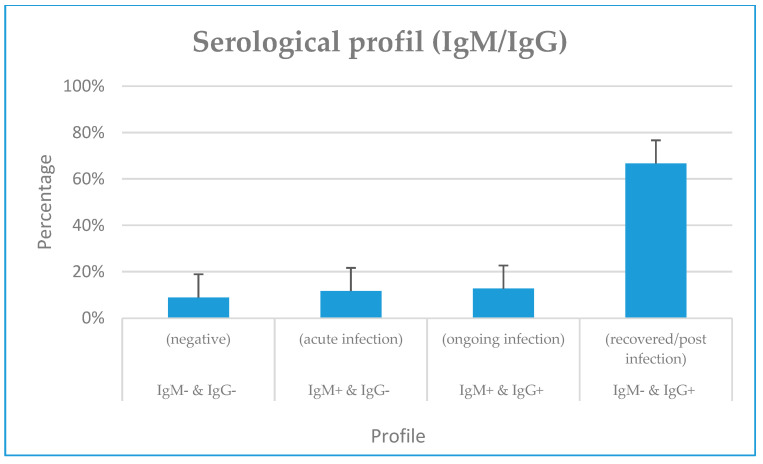
Overall IgM/IgG serological profile of the study participants.

**Table 1 viruses-15-00407-t001:** Sociodemographic Characteristics of the Study population.

Variables		*n*	Percentage (%)
Age range (year)	18–21	81	27.84
	21–24	120	41.24
	24–28	90	30.93
Gender	Male	138	47.42
	Female	153	52.58
Location	Yaounde	93	31.96
	Bangangte	198	68.04
Clinical status	Symptomatic	57	19.59
	Asymptomatic	234	80.41
Comorbidities	Yes	12	4.12
	No	279	95.88
Case-contact	Yes	56	19.24
	No	235	80.76
Nomadic ^1^	Yes	20	6.87
	No	271	93.13
Treatment	Yes	4	1.37
	No	287	98.63

^1^ Nomadic: Participant who had moved out of their town in the past two weeks prior to enrolment in the study; ^2^ Sedentary: No migration experienced out of town within the past two weeks.

**Table 2 viruses-15-00407-t002:** Risk factors of infection with COVID-19 in the Study Population.

Variables	N	Positive PCR *n*(%)	ORa	95%CI	*p*-Value
Age range (year)					
[18–21]	81	15 (18.52)	Ref		
[21–24]	120	31 (25.83)	1.81	0.8–3.7	0.11
[24–28]	90	16 (17.78)	1.09	0.4–2.5	0.83
Sexe					
Female	153	44 (28.76)	Ref		
Male	138	18 (13.04)	2.21	1.5–4.5	0.0007
Location					
Bangangte	198	50 (25.25)	Ref		
Yaounde	93	12 (12.90)	1.95	1.2–3.8	0.01
Clinical status					
Asymptomatic	234	53 (22.65)	Ref		
Symptomatic	57	9 (15.79)	0.64	0.3–1.4	0.2
Comorbidities					
No	279	58 (20.79)	Ref		
Yes	12	4 (33.33)	1.14	0.3–4.3	0.84
Case-contact					
No	235	52 (22.13)	Ref		
Yes	56	10 (17.86)	0.61	0.3–1.4	0.25
Nomadic					
No	271	57 (21.03)	Ref		
Yes	20	5 (25)	1.82	0.5–6.1	0.33
Treatment					
No	287	61 (21.25)	Ref		
Yes	4	1 (25)	1.70	0.15–18.7	0.66

PCR = Polymerase Chain Reaction; ORa = Adjusted Odds Ratio.

**Table 3 viruses-15-00407-t003:** IgM/IgG profile by sociodemographic and clinical characteristics.

Variables	Total (N)	IgM+&IgG+ *n*(%)	*p*-Value	IgM+&IgG− *n*(%)	*p*-Value	IgM−&IgG+ *n*(%)	*p*-Value
Age range (year)							
[18–21]	81	8 (9.88)	Ref	4 (4.94)	Ref	13 (16.05)	Ref
[21–24]	120	16 (13.33)	0.26	11 (9.17)	0.5	13 (10.83)	0.31
[24–28]	90	10 (11.11)	0.61	11 (12.22)	0.21	11 (12.22)	0.63
Gender							
Female	153	24 (15.69)	Ref	12 (7.84)	Ref	19 (12.42)	Ref
Male	138	10 (7.25)	0.01	14 (10.14)	0.77	18 (13.04)	0.92
Location							
Bangangte	198	28 (14.14)	Ref	21 (10.61)		23 (11.62 %)	Ref
Yaounde	93	6 (4.45)	0.05	5 (5.38)	0.11	14 (15.05 %)	0.74
Clinical status							
Asymptomatic	234	30 (12.82)	Ref	21 (8.97)	Ref	28 (11.97)	Ref
Symptomatic	57	4 (7.02)	0.15	5 (8.77)	0.67	9 (15.79)	0.6
Comorbidities							
No	279	34 (12.19)	Ref	23 (8.24)	Ref	34 (12.19)	Ref
Yes	12	0 (0.00)	0.97	3 (25.00)	0.12	3 (25.00)	0.11
Case-contact							
No	235	30 (12.77)	Ref	19 (8.09)	Ref	32 (13.62)	Ref
Yes	56	4 (7.14)	0.05	7 (12.50)	0.9	5 (8.93)	0.18
Nomad							
No	271	30 (11.07)	Ref	23 (8.49)	Ref	34 (12.55)	Ref
Yes	20	4 (20.00)	0.01	3 (15.00)	0.3	3 (15.00)	0.21
Treatment							
No	287	33 (11.50)	Ref	25 (8.71)	Ref	37 (12.89)	Ref
Yes	4	1 (25.00)	0.22	1 (25.00)	0.2	0 (0.00)	0.98

**Table 4 viruses-15-00407-t004:** Serological profile (IgM/IgG) and SARS-CoV-2 RNA detection.

Profil Sérologique	N	PCR		OR	IC 95%	*p*-Value
		Negative *n* (%)	Positive *n* (%)			
IgM− & IgG−	194	180 (92.98)	14 (7.22)	Ref		
*(negative)*						
IgM+ & IgG−	26	7 (26.92)	19 (73.08)	34.9	[12.5–97]	<0.0001
*(acute infection)*						
IgM+ & IgG+	34	6 (17.65)	28 (82.35)	60	[21.3–169]	<0.0001
*(ongoing infection)*						
IgM− & IgG+	37	36 (97.3)	1 (2.7)	0.36	[0.05–2.8]	0.33
*(recovered/post infection)*						

key: PCR = Polymerase chain reaction; OR = Odds-ratio; IC = confidence Interval; Inf = Infection.

## Data Availability

All data generated by the study are included in the manuscript.

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
