# Peer review of "Comparative Performance of Serological (IgM/IgG) and Molecular Testing (RT-PCR) of COVID-19 in Three Private Universities in Cameroon during the Pandemic"

_viruses, 2023, doi:10.3390/v15020407_

Round 1

Reviewer 1 Report

While I believe that this manuscript has merit, there are several aspects that need to be addressed in order to improve the quality of the manuscript.

GENERAL COMMENTS
Repetitive/Redundant statements, e.g. "SARS-CoV-2 is a 30 kb enveloped virus" (line 63) and again "30,000 nucleotides" (line 65); the latter part of lines 88 and 89; etc. Please revise such statements and others in the manuscript.
Incorrect wording, e.g. "pseudo symptoms" (line 73), "cytokine oracle" (line 79), "viral profiling" (line 332), serological scars (line 342), etc. Please stick to standard naming conventions and nomenclature used so that readers understand the concepts being described by the authors and ensure coherence within the relevant fields.
Inconsistent referral to universities/locations: the authors refer to the universities and their locations interchangeably without first establishing a consistent naming convention. This creates confusion, especially for those who aren't familiar with the geography of Cameroon. It's also confusing because the authors initially say "three locations of Cameroon (West, Littoral, and Center)" (lines 37-38) but then later merely refer to two locations, i.e. "Yaounde" and "Bangangte" halfway through the remainder of the manuscript. It is important that the authors clearly state which regions were used, establish a naming convention, and consistently use it throughout the manuscript.
Leave the reader to draw their own conclusion: at times the authors don't fully demonstrate/explain the importance of their results, leaving the reader to draw their own conclusions. It's important that the authors clearly show the reader how their results relate to other studies.

TITLE
I suggest the authors revise the title since RT-qPCR does not constitute "virological markers" and "immunological markers" entail more than just "IgM/IgG" measurements. Using these terms without providing the necessary measurements to support them is misleading.

ABSTRACT
• Line 33: The authors talk about COVID-19 being "unreported in resource-limited settings". While this is largely true for many regions of the world, why do the authors mention this while their study investigates COVID-19 in "private universities". It seems a bit contradictory, and I suggest the authors rephrase/clarify this.
• Line 38: The authors never stated why they chose these three locations/universities to conduct their study. Please include the rationale for using these universities to conduct the study in the Methodology section.
• Line 46: It is unclear why the authors reported ONLY the P-value here without any accompanying data to provide some context.
• Line 50: It is unclear what the IgM status is in the sentence "Lastly, 7.22% (14/194) with IgM/IgG- had a positive PCR." Please include the status of both IgM and IgG at all times to avoid confusion.
• Lines 53-55: why do the authors appear to advocate for the use of more sensitive molecular assays (which are generally more expensive), such as PCR, when they spoke about "resource-limited settings" in line 33? Please clarify.

INTRODUCTION
While much of the background provided is largely correct, the introduction section largely does not give much background relevant to the aspects of the study, such as viral spread, seroprevalence/seroconversion, or rate of infectivity in similar university settings. These would be more relevant and provide the reader with a better understanding of the context of the current study.
• Line 61: SARS-CoV-2 is not a variant of the Coronaviridae family. It is a new virus. Please correct this.
• Lines 72-73: where is the reference to support "COVID-19 disease is asymptomatic in 85% of cases"? Reference 4 (Rabi et al., 2020) does not report asymptomatic cases as high as 85% - please update this. Many studies throughout the course of the pandemic reported asymptomatic cases to be much lower than 85%. 

MATERIALS and METHODS
There appear to be quite a few pieces of information missing from this section: e.g. at which time points were the nasopharyngeal swabs done? Blood sampling?
• The RT-qPCR conditions and how the assay was conducted are missing - please include this
• Line 109: I don't quite agree that this study is both a cross-sectional and prospective study. Typically, cross-sectional studies identify disease prevalence, whereas prospective studies look at disease progression over time. Since the authors did not provide any information on the time points at which sampling was done and whether participants were monitored over time, it is unclear whether this study is truly cross-sectional or prospective. However, they cannot be both since the former identifies prevalence and the latter incidence.
• Lines 110-113: please clarify the regions in which each university is located using a consistent naming convention.
• Line 124: how do the authors differentiate between COVID-19 and other respiratory conditions that can cause similar symptoms, if they use "at least one" of the symptoms listed in the questionnaire? Also, what were the inclusion and exclusion criteria for participants? Please include this in this section.
• Is the questionnaire used in this study a standardised questionnaire or was it specifically designed only for this study? It is unclear how this questionnaire was conducted: did participants complete it on their own (self-reported) or did they complete it in a healthcare setting with a health-care professional?
• Line 139: what were the eligibility criteria for participants?
• Lines 184 and 186: inconsistent naming when referring to the COVID-19 antibody testing - is it "Abbot" or "Panbio"? Please choose a naming convention and stick to it.

RESULTS:
• Lines 219-220: why are participants suddenly grouped as being from "Bangangte" or "Yaounde", instead of from the three universities as indicated in the Method section? This creates confusion.
• Lines 223-224: it is unclear what "sedentary" and "nomadic" mean in the context of this study. Please include the necessary contextual definitions in the manuscript.
• Table 1: if the questionnaire was self-reported, how can the authors be certain that all participants were completely honest and transparent in admitting whether they had case-contact?
• Lines 235-237: the authors report an odds-ratio of 1.95 between Bangangte and Yaounde, suggesting that persons from Bangangte are 1.95 times more likely to have COVID-19 than those from Yaounde. However, since almost 70% of the participants were from Bangangte (Table 1), I don't agree that this odds-ratio accurately reflects the likelihood of a person being infected between the two cities. Similarly, more females had a positive PCR but there were also more females in the study than males.
• Table 2: did the authors adjust/account for any potential confounding factors? 
• Lines 246-248: I don't agree that the % seropositivity of IgM+/IgG+ should be included in BOTH IgM and IgG groups. It would be clearer to present the groups in their own individual groups as was done in Figure 2. And, IgM-/IgG+ is not "cured" or "chronic". I have not seen studies to suggest chronic COVID-19 due to latent infection. I feel that this group can be more accurately described as "recovered".

DISCUSSION
This section should also include comparing similarly conducted studies and studies in the same context. It would add a lot more value to this manuscript if the authors discussed their results in the context of and also compared it to the results of other similar studies.
• Lines 309-311: it appears as though the authors are referring to the importance of neutralising antibodies and the potential for reinfection as their main argument, but their study didn't investigate neutralising antibodies or reinfection. Can the authors explain how their results are rerelevant to neutralising antibodies and reinfection?
• Lines 326-327: this statement appears to contradict the sensitivity (98.2%) and specificity (99.4%) of the test kit used by the authors at lines 190-191. Could the authors please clarify?
• Lines 334-337: how does the authors' argument here speak to resource-limited settings as the authors referred to in lines 33-34? Techniques such as RT-PCR and laboratory-based serology, while more accurate, are generally also more expensive and require more "specialist" personnel, making them difficult to implement in resource-limited settings.
• Line 341: I caution the authors against the use of preprints as peer-reviewed evidence. I would advise the authors to include that the study they are referring to is currently a preprint.

Author Response

GENERAL COMMENTS
• Repetitive/Redundant statements, e.g. "SARS-CoV-2 is a 30 kb enveloped virus" (line 63) and again "30,000 nucleotides" (line 65); the latter part of lines 88 and 89; etc. Please revise such statements and others in the manuscript.

Answer: Thanks dear reviewer, we revised it

  • Incorrect wording, e.g. "pseudo symptoms" (line 73), "cytokine oracle" (line 79), "viral profiling" (line 332), serological scars (line 342), etc. Please stick to standard naming conventions and nomenclature used so that readers understand the concepts being described by the authors and ensure coherence within the relevant fields.

Answer: Thanks dear reviewer, we corrected them as requested

  • Inconsistent referral to universities/locations: the authors refer to the universities and their locations interchangeably without first establishing a consistent naming convention. This creates confusion, especially for those who aren't familiar with the geography of Cameroon. It's also confusing because the authors initially say "three locations of Cameroon (West, Littoral, and Center)" (lines 37-38) but then later merely refer to two locations, i.e. "Yaounde" and "Bangangte" halfway through the remainder of the manuscript. It is important that the authors clearly state which regions were used, establish a naming convention, and consistently use it throughout the manuscript.

Answer: Thanks dear reviewer, we did it.

  • Leave the reader to draw their own conclusion: at times the authors don't fully demonstrate/explain the importance of their results, leaving the reader to draw their own conclusions. It's important that the authors clearly show the reader how their results relate to other studies.

Answer: Thanks dear reviewer

TITLE
I suggest the authors revise the title since RT-qPCR does not constitute "virological markers" and "immunological markers" entail more than just "IgM/IgG" measurements. Using these terms without providing the necessary measurements to support them is misleading.

Answer: Thanks dear reviewer, we revised it as follow. “Profile of serological (IgM/IgG) and virological markers of COVID-19 in some private universities in Cameroon during the pandemic.”

ABSTRACT
• Line 33: The authors talk about COVID-19 being "unreported in resource-limited settings". While this is largely true for many regions of the world, why do the authors mention this while their study investigates COVID-19 in "private universities". It seems a bit contradictory, and I suggest the authors rephrase/clarify this.

Answer: Thanks dear reviewer, we rephrased it

  • Line 38: The authors never stated why they chose these three locations/universities to conduct their study. Please include the rationale for using these universities to conduct the study in the Methodology section.

Answer: Thanks dear reviewer, we did it in the methodology as requested

  • Line 46: It is unclear why the authors reported ONLY the P-value here without any accompanying data to provide some context.

Answer: Dear reviewer, more details have been given in the results section

  • Line 50: It is unclear what the IgMstatus is in the sentence "Lastly, 7.22% (14/194) with IgM/IgG- had a positive PCR." Please include the status of both IgM and IgG at all times to avoid confusion.

Answer: Corrected, thanks dear reviewer

  • Lines 53-55: why do the authors appear to advocate for the use of more sensitive molecular assays (which are generally more expensive), such as PCR, when they spoke about "resource-limited settings" in line 33? Please clarify.

Answer: Dear reviewer, during the COVID-19 pandemic, molecular assays for RNA detection have been made available even in low and middle-income settings. Given that they are more expensive and thus, not widely accessible, an extrinsic goal of this study is to help to put in place simple/efficient diagnostic strategies (algorithms) to support the pandemic response. And this disparity between IgG/IgM and viral profile supports prioritizing viral assays (nucleic acid or antigen) for diagnosis and antibody screening for sero-surveys. Moreover, a medical recommendation for a test by a Physician is free in Cameroon  

INTRODUCTION
While much of the background provided is largely correct, the introduction section largely does not give much background relevant to the aspects of the study, such as viral spread, seroprevalence/seroconversion, or rate of infectivity in similar university settings. These would be more relevant and provide the reader with a better understanding of the context of the current study.

Answer:  We revised the introduction accordingly, thanks dear reviewer

  • Line 61: SARS-CoV-2 is not a variant of the Coronaviridae family. It is a new virus. Please correct this.

Answer: Corrected, thanks dear reviewer

  • Lines 72-73: where is the reference to support "COVID-19 disease is asymptomatic in 85% of cases"? Reference 4 (Rabi et al., 2020) does not report asymptomatic cases as high as 85% - please update this. Many studies throughout the course of the pandemic reported asymptomatic cases to be much lower than 85%. 

Answer: Corrected, thanks dear reviewer

MATERIALS and METHODS
There appear to be quite a few pieces of information missing from this section: e.g. at which time points were the nasopharyngeal swabs done? Blood sampling?

Answer: Dear reviewer, this was a cross sectional study. We collected samples in three different study locations and performed analyses once.

  • The RT-qPCR conditions and how the assay was conducted are missing - please include this

Answer: Included dear reviewer. Thanks

  • Line 109: I don't quite agree that this study is both a cross-sectional and prospective study. Typically, cross-sectional studies identify disease prevalence, whereas prospective studies look at disease progression over time. Since the authors did not provide any information on the time points at which sampling was done and whether participants were monitored over time, it is unclear whether this study is truly cross-sectional or prospective. However, they cannot be both since the former identifies prevalence and the latter incidence.

Answer: Dear reviewer, we simply removed the word prospective to avoid misunderstandings

  • Lines 110-113: please clarify the regions in which each university is located using a consistent naming convention.

Answer: Done dear reviewer

  • Line 124: how do the authors differentiate between COVID-19 and other respiratory conditions that can cause similar symptoms, if they use "at least one" of the symptoms listed in the questionnaire?

Also, what were the inclusion and exclusion criteria for participants? Please include this in this section.

Answer: we reformulate that section dear reviewer, thanks

  • Is the questionnaire used in this study a standardised questionnaire or was it specifically designed only for this study? It is unclear how this questionnaire was conducted: did participants complete it on their own (self-reported) or did they complete it in a healthcare setting with a health-care professional?

Answer: Dear reviewer, as the study was conducted at the virology laboratory of the Ministry of Scientific Research and Innovation (CREMER/IMPM/MINRESI-IRD/UMI233) we just adapted the standardized questionnaire that has approved nationwide to our study. As mentioned in the methodology, each participant completed the questionnaire in presence of a research team member.

  • Line 139: what were the eligibility criteria for participants?

Answer: Dear reviewer, we gave more details in the methology as requested.

  • Lines 184 and 186: inconsistent naming when referring to the COVID-19 antibody testing - is it "Abbot" or "Panbio"? Please choose a naming convention and stick to it.

Answer: Ok dear reviewer.

RESULTS:
• Lines 219-220: why are participants suddenly grouped as being from "Bangangte" or "Yaounde", instead of from the three universities as indicated in the Method section? This creates confusion.

Answer: Thanks dear reviewer, we have corrected. As we just clarified in the methodology, one private university was located in Bangangte, in the West region Cameroon (UdM-Bangangte) and the two others in Yaounde, in the central region Cameroon (IUEs/INSAM-Yaounde and ISSBA-Yaounde)

  • Lines 223-224: it is unclear what "sedentary" and "nomadic" mean in the context of this study. Please include the necessary contextual definitions in the manuscript.

Answer: thanks dear reviewer, we have corrected.

  • Table 1: if the questionnaire was self-reported, how can the authors be certain that all participants were completely honest and transparent in admitting whether they had case-contact?

Answer: No dear reviewer, each questionnaire was filled in the presence of a research team member.

  • Lines 235-237: the authors report an odds-ratio of 1.95 between Bangangte and Yaounde, suggesting that persons from Bangangte are 1.95 times more likely to have COVID-19 than those from Yaounde. However, since almost 70% of the participants were from Bangangte (Table 1), I don't agree that this odds-ratio accurately reflects the likelihood of a person being infected between the two cities. Similarly, more females had a positive PCR but there were also more females in the study than males.
    • Table 2: did the authors adjust/account for any potential confounding factors? 

Answer:  Dear Reviewer, as you can see in the table, the reported (ORa) has been adjusted to all other variables to reduce the effect of any potential confounding factors.

  • Lines 246-248: I don't agree that the % seropositivity of IgM+/IgG+should be included in BOTH IgM and IgG groups. It would be clearer to present the groups in their own individual groups as was done in Figure 2.

Answer: Thanks dear reviewer, we intended to highlight the overall proportion of IgM+ on one side and IgG+ on the other side.

And, IgM-/IgG+ is not "cured" or "chronic". I have not seen studies to suggest chronic COVID-19 due to latent infection. I feel that this group can be more accurately described as "recovered".

 Answer: thanks dear reviewer, we described them as recovered or post-inflectional immunity

DISCUSSION
This section should also include comparing similarly conducted studies and studies in the same context. It would add a lot more value to this manuscript if the authors discussed their results in the context of and also compared it to the results of other similar studies.
• Lines 309-311: it appears as though the authors are referring to the importance of neutralising antibodies and the potential for reinfection as their main argument, but their study didn't investigate neutralising antibodies or reinfection. Can the authors explain how their results are rerelevant to neutralising antibodies and reinfection?

Answer: thanks dear reviewer, we reformulated it
• Lines 326-327: this statement appears to contradict the sensitivity (98.2%) and specificity (99.4%) of the test kit used by the authors at lines 190-191. Could the authors please clarify?

Answer: thanks dear reviewer for that remark, it was an oversight that has been corrected

  • Lines 334-337: how does the authors' argument here speak to resource-limited settings as the authors referred to in lines 33-34? Techniques such as RT-PCR and laboratory-based serology, while more accurate, are generally also more expensive and require more "specialist" personnel, making them difficult to implement in resource-limited settings.

Answer: Dear reviewer, As COVID-19 emerged in early 2020, invaluable efforts were made by many states/governments in low and middle-income countries with the help of WHO to meet the standard requirements for the efficient diagnostic of the virus including techniques such as RT-PCR and laboratory-based serology and thus, better respond to the pandemic. It means things rely somewhere on political will, and as a scientist, we just need to provide evidence that will lead to good decision-making.

  • Line 341: I caution the authors against the use of preprints as peer-reviewed evidence. I would advise the authors to include that the study they are referring to is currently a preprint.

Answer: Ok dear reviewer

Submission Date

22 December 2022

Date of this review

30 Dec 2022 22:22:40

Reviewer 2 Report

The manuscript is well written and reading flows well. The methodology is suitable too. I have a few comments for the authors to improve the manuscript, after which I believe the manuscript can be published.

1. Abstract: it is too long, I believe the maximum words for abstracts at mdpi journals is 250 words? The conclusion section in the abstract for example could have only the sentence from lines 53-55. Therefore I suggest shortening the abstract in order to have 250 words.

2. Lines 61-62: reformulate the sentence without the word "variant", as it might be confusing for the readers due to the existence of SARS-CoV-2 VoCs... it could be something like "the surge of a new virus from the Coronaviridae family".

3. Lines 86-87. change sentence to something like "new variants that are more contagious and virulent"

4. Lines 88-89. last sentence can be removed.

5. Line 91. add "due to COVID-19" at the end of this sentence.

6. Lines 90-97. please update these numbers according to the more recent numbers released in WHO reports.

7.  Lines 98-100. is that sentence true? do we really know LITTLE about COVID-19 pathogenesis and immune response by now?

8. Lines 101-102. I think the objective of this research should be reformulated. IgG/IgM seroprevalence does not tell us anything about the immunological response per se. It tells us our bodies are producing antibodies against the virus and that's it. Perhaps you should say instead that the aim was to investigate the seroprevalence of IgG and IgM antibodies.

9. Line 124. change "s/he" for "he/she".

10. Line 181. Change to "Detection of SARS-CoV-2 RNA and IgM/IgM antibodies"

11. Line 241. Figure 1 might be removed as it does not bring any new information that was not already mentioned in text.

12. Line 260. Figure 2 should be reformulated, in order for its layout to look a little more suitable for publication? As it is right now it looks like a straight-out-of-excel graph...

13. Line 265. change to "and PCR results for SARS-CoV-2 RNA detection"

14. Line 267. substitute "immunological profile" for "serological profile" throughout the manuscript.

15. The discussion overall is nice, but they should discuss a little the results of similar studies with similar objectives published thus far. I leave here a few examples:

https://www.ncbi.nlm.nih.gov/pmc/articles/PMC7541966/

https://www.ssph-journal.org/articles/10.3389/ijph.2022.1604548/full

https://www.ncbi.nlm.nih.gov/pmc/articles/PMC8544125/

https://www.frontiersin.org/articles/10.3389/fpubh.2022.967447/full

https://www.nature.com/articles/s41598-022-09215-8

Author Response

The manuscript is well written and reading flows well. The methodology is suitable too. I have a few comments for the authors to improve the manuscript, after which I believe the manuscript can be published.

  1. Abstract: it is too long, I believe the maximum words for abstracts at mdpi journals is 250 words? The conclusion section in the abstract for example could have only the sentence from lines 53-55. Therefore I suggest shortening the abstract in order to have 250 words.

Answer: Thanks dear reviewer, we shorten it

  1. Lines 61-62: reformulate the sentence without the word "variant", as it might be confusing for the readers due to the existence of SARS-CoV-2 VoCs... it could be something like "the surge of a new virus from the Coronaviridae family".

Answer: Corrected dear reviewer, Thanks.

  1. Lines 86-87. change sentence to something like "new variants that are more contagious and virulent"

Answer: Corrected dear reviewer, Thanks

  1. Lines 88-89. last sentence can be removed.

Answer: Thanks dear reviewer

  1. Line 91. add "due to COVID-19" at the end of this sentence.

Answer: Done, thanks dear reviewer

  1. Lines 90-97. please update these numbers according to the more recent numbers released in WHO reports.

Answer: Done dear reviewer

  1. Lines 98-100. is that sentence true? do we really know LITTLE about COVID-19 pathogenesis and immune response by now?

Answer: We reformulated it dear reviewer, thanks

  1. Lines 101-102. I think the objective of this research should be reformulated. IgG/IgM seroprevalence does not tell us anything about the immunological response per se. It tells us our bodies are producing antibodies against the virus and that's it. Perhaps you should say instead that the aim was to investigate the seroprevalence of IgG and IgM antibodies.

Answer: We reformulated it dear reviewer, thanks

  1. Line 124. change "s/he" for "he/she".

Answer: Corrected, thanks dear reviewer

  1. Line 181. Change to "Detection of SARS-CoV-2 RNA and IgM/IgM antibodies"

Answer: Corrected, thanks dear reviewer

  1. Line 241. Figure 1 might be removed as it does not bring any new information that was not already mentioned in text.

Answer: Done dear reviewer

  1. Line 260. Figure 2 should be reformulated, in order for its layout to look a little more suitable for publication? As it is right now it looks like a straight-out-of-excel graph...

Answer: Done dear reviewer

  1. Line 265. change to "and PCR results for SARS-CoV-2 RNA detection"

Answer: Done dear reviewer

  1. Line 267. substitute "immunological profile" for "serological profile" throughout the manuscript.

Answer: Done dear reviewer

  1. The discussion overall is nice, but they should discuss a little the results of similar studies with similar objectives published thus far. I leave here a few examples:

https://www.ncbi.nlm.nih.gov/pmc/articles/PMC7541966/

https://www.ssph-journal.org/articles/10.3389/ijph.2022.1604548/full

https://www.ncbi.nlm.nih.gov/pmc/articles/PMC8544125/

https://www.frontiersin.org/articles/10.3389/fpubh.2022.967447/full

https://www.nature.com/articles/s41598-022-09215-8

Answer: Thanks dear reviewer for those interesting studies

Submission Date

22 December 2022

Date of this review

04 Jan 2023 15:50:54

Reviewer 3 Report

Text must be rechecked due to some technical mistakes, e.g.

row 222  comorbidity (4.1295.88%). - incorrect data presentation

I strongly recommend to authors use the term "postinfectional or porstvaccination immunity" when they describe IgG+/IgM- serological profile instead of "immunological markers of immunization/chronic infection". As we know stlii there are no evidence of COVID-19 chronization so isn't correct to use such definition regarding carriers of IgG antibodies.

For me still not clear such great disproportion of females positivity vs COVID-19 prevalence amongst males. It will be nice to clarify or comment this topic if it is possible.

Author Response

Text must be rechecked due to some technical mistakes, e.g.

Answer: Dear reviewer, we deeply revised the text as requested.

row 222  comorbidity (4.1295.88%). - incorrect data presentation

 Answer: Corrected, dear reviewer

I strongly recommend to authors use the term "postinfectional or porstvaccination immunity" when they describe IgG+/IgM- serological profile instead of "immunological markers of immunization/chronic infection". As we know stlii there are no evidence of COVID-19 chronization so isn't correct to use such definition regarding carriers of IgG antibodies.

Answer: Thanks very much for that comment, we chose postinfectional immunity as none of our participants was vaccinated.

For me still not clear such great disproportion of females positivity vs COVID-19 prevalence amongst males. It will be nice to clarify or comment this topic if it is possible.

Answer: We further explained it dear reviewer

Submission Date

22 December 2022

Date of this review

02 Jan 2023 16:17:22

Round 2

Reviewer 1 Report

While the authors addressed previous suggestions, there are still some improvements that can be made.

TITLE
The authors have amended "immunological markers" to "serological markers", which is more accurate and appropriate for the data generated by this study. However, this study did not assess or generate any "virological markers" as the title suggests. The RT-qPCR alone from this study does not constitute "virological markers". I suggest the authors rephrase "virological markers" to a description more befitting of the RT-qPCR data.

Line 52: Typo at "80.41% (234/2691)"

Line 91: It would be more appropriate to refer to the 7774 patients as "recovered" instead of "cured".

Line 124: Typo at "privates"

Subheading "4. COVID-19 Testing": I believe a description of the RNA isolation is missing from the methodology. I think it would benefit the manuscript to include the RNA isolation protocol under this subheading.

Line 264: The latter part of line 264 does not read well, "almost all our most" - please rephrase this.

Lines 363-365: This statement lacks referencing. There are no papers referenced, despite the authors referring to "experience earned from the kinetics of antibody response from other viral infections".

Author Response

TITLE
The authors have amended "immunological markers" to "serological markers", which is more accurate and appropriate for the data generated by this study. However, this study did not assess or generate any "virological markers" as the title suggests. The RT-qPCR alone from this study does not constitute "virological markers". I suggest the authors rephrase "virological markers" to a description more befitting of the RT-qPCR data.

Answer: Thanks dear reviewer, The Title has been once more reformatted “Comparative performance of serological (IgM/IgG) and molecular testing (RT-PCR) of COVID-19 in 3 private universities in Cameroon during the pandemic”

Line 52: Typo at "80.41% (234/2691)"

Answer: Corrected dear reviewer, Thanks

Line 91: It would be more appropriate to refer to the 7774 patients as "recovered" instead of "cured".

Answer: Corrected dear reviewer, Thanks

Line 124: Typo at "privates"

Answer: Corrected dear reviewer, Thanks

Subheading "4. COVID-19 Testing": I believe a description of the RNA isolation is missing from the methodology. I think it would benefit the manuscript to include the RNA isolation protocol under this subheading.

Answer: included dear reviewer, Thanks.

Line 264: The latter part of line 264 does not read well, "almost all our most" - please rephrase this.

Answer: Corrected dear reviewer, Thanks

Lines 363-365: This statement lacks referencing. There are no papers referenced, despite the authors referring to "experience earned from the kinetics of antibody response from other viral infections".

Answer: Added dear reviewer, Thanks.

Submission Date

22 December 2022

Date of this review

23 Jan 2023 21:48:26

Reviewer 2 Report

I believe the authors have addressed all my comments properly. If this is the case regarding the other reviewer's comments, I believe the manuscript can be published after minor reviews in english grammar/spell check.

Author Response

I believe the authors have addressed all my comments properly. If this is the case regarding the other reviewer's comments, I believe the manuscript can be published after minor reviews in english grammar/spell check.

Answer: Thanks very much once more for your thoughtful suggestions and insights dear reviewer